# Is Programming by Example Solved by LLMs?

**Wen-Ding Li**
Cornell University
`wl678@cornell.edu`

**Kevin Ellis**
Cornell University
`kellis@cornell.edu`

## Abstract

Programming-by-Examples (PBE) aims to generate an algorithm from input-output examples. Such systems are practically and theoretically important: from an end-user perspective, they are deployed to millions of people, and from an AI perspective, PBE corresponds to a very general form of few-shot inductive inference. Given the success of Large Language Models (LLMs) in code-generation tasks, we investigate here the extent to which LLMs can be said to have 'solved' PBE. We experiment on classic domains such as lists and strings, and an uncommon graphics programming domain not well represented in typical pretraining data. We find that pretrained models are not effective at PBE, but that they can be fine-tuned for much higher performance, provided the test problems are in-distribution. We analyze empirically what causes these models to succeed and fail, and take steps toward understanding how to achieve better out-of-distribution generalization. Collectively these results suggest that LLMs make strong progress toward solving the typical suite of PBE tasks, potentially increasing the flexibility and applicability of PBE systems, while also identifying ways in which LLMs still fall short.

## 1 Introduction

Programming-by-Example (PBE) systems solve a challenging task: Given input-output examples of a hidden algorithm, they seek to construct the source code of the underlying function [1, 2]. PBE is deployed to millions of users [3, 4, 5, 6], lies near the heart of core AI challenges [7, 8, 9, 10], and is a qualitatively different problem from the bulk of recent work on LLM code generation, because rather than generate source code from natural language [11], PBE is instead fundamentally about few-shot inductive inference: Given a handful of examples, inferring the program that will generalize to new inputs, or which captures the true latent regularity, *without* relying on natural-language guidance.

We investigate here the extent to which large language models pretrained on source code can solve PBE. If they can, this unlocks the ability to do PBE in general-purpose Turing complete languages like Python, unlike the restricted domain-specific languages which have so far dominated PBE [4, 12, 13, 14, i.a.], thereby increasing the scope and power of this paradigm. If LLMs cannot perform PBE, then this highlights a deficit of inductive reasoning and problem solving, and suggests LLMs lean too heavily on natural language cues to generate code.

We find that pretrained and instruction-tuned models serve as poor PBE systems, a finding also supported by recent work [15, 16, 12, 17]. But our investigation further finds that LLMs can be fine-tuned for significantly higher performance, provided they are not asked to generalize far beyond the fine-tuning data. To address this failure of generalization we give an algorithm for taking a small unlabeled dataset of problems and adapting the LLM to it, which we find narrows this domain gap.

The resulting recipe allows PBE over Turing-complete languages across three qualitatively different domains (Fig. 1): algorithms on vectors of numbers, string manipulation macros, and graphics programs in LOGO/Turtle. In every case, our final model is at least as effective as custom symbolic search algorithms operating over domain-specific languages, and surpasses powerful closed-source

38th Conference on Neural Information Processing Systems (NeurIPS 2024).

models such as GPT4 [18]. We also find that the resulting system can cover a broader scope of problems than classic symbolic methods, owing to the use of a Turing-complete language, which, at least theoretically, allows learning any computable function.

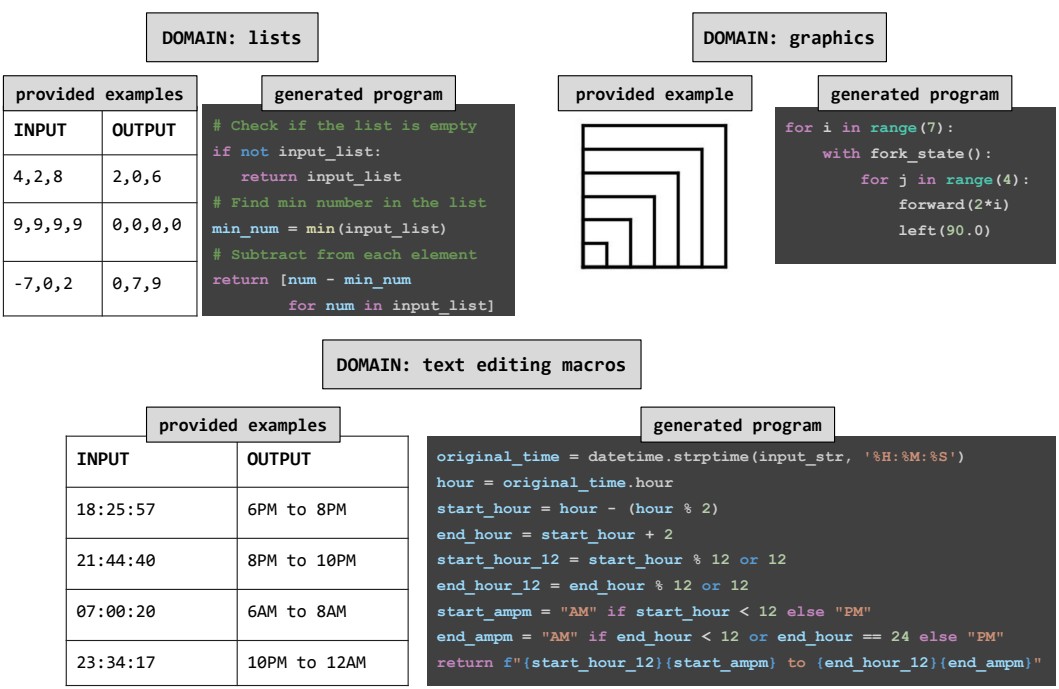

Figure 1: Domains, including standard ones that resemble programs found in pretraining data, as well as a less common graphics domain, which is likely less represented in LLM pretraining data.

## 2   Background

**Programming by Example** considers synthesizing a program $\rho$ given a vector of inputs $X$ and corresponding outputs $Y$. Typically the program is expected to exactly fit the provided examples, $\rho(X_i) = Y_i, \forall i$, where $i$ indexes examples. The program $\rho$ is drawn from a (potentially infinite) language $\mathcal{L}$. Typically $\mathcal{L}$ is a domain-specific language designed for a specific PBE system, not a general-purpose programming language. For example, the PBE system FlashFill synthesizes string manipulation macros designed to automate common spreadsheet edits [2]. FlashFill's domain-specific language $\mathcal{L}$ includes commonly occurring regular expressions, together with string slicing and concatenation, and restricted forms of loops. The language $\mathcal{L}$ is also designed to allow polynomial-time construction of programs consistent with input-output examples. FlashFill's goal, like most PBE systems, is to generalize to hold-out test cases: inputs $X'$ with (hidden) target outputs $Y'$.

$$\text{Pick a program } \rho \text{ from } \{\rho \in \mathcal{L} \; : \; \rho(X_i) = \rho(Y_i), \forall i\}. \quad \text{Succeed if } \rho(X'_j) = \rho(Y'_j), \forall j \quad (1)$$

In its simplest forms, PBE can be accomplished by guess-and-check enumeration until a program is found that is consistent with the examples. Although there exist more sophisticated search algorithms, including those accelerated by neural guidance [19, 20, 21, 22], a key enabler of practical PBE systems is the design of a carefully restricted domain-specific language $\mathcal{L}$. The domain-specific language effectively hardcodes symbolic knowledge, focusing the system on what programs the human engineer thinks are most promising, but at the expense of the wider set of computable functions expressible in general-purpose languages.

The PBE setup covers other cases as well, such as sequence extrapolation (the inputs are indices into the sequence), as well as data compression (the input is null, and the data is compressed by synthesizing a program that reproduces the output data). Therefore, a truly general solution to PBE—one which could express its solutions in general purpose programming languages, and cover most

practically relevant problems—would be broadly applicable to many inductive inference problems, a point that has been long appreciated [9].

**LLMs for solving programming problems** have been recently very successful [11, 23, 24, 25, 26]. These systems typically input a prompt describing a problem in natural language, then sample candidate programs, and optionally filter those samples by checking them against input-output test cases, with the goal of passing holdout tests:

Draw $\rho_k \sim p_{\text{LM}}(\cdot|\texttt{prompt})$. Pick a $\rho \in \{\rho_k \ : \ \rho_k(X_i) = \rho_k(Y_i), \forall i\}$. Success: $\rho(X'_j) = \rho(Y'_j), \forall j$

Unlike PBE, the primary driver of program generation is a natural language prompt, although input-outputs may also be in the prompt [27, 28]. Recent work using LLMs to synthesize programs solely from examples has either obtained negative results [16, 12], or focused on simple and/or nonstandard problems [29, 30, 31], leaving the extent to which PBE is 'solved' by LLMs an open question.

## 3 Methods

**Basic prompting** is the most straightforward way of performing PBE with a pre-trained model: Given input-output examples $(X, Y)$ a prompt is constructed and $K$ programs are generated. Programs are filtered by the I/O examples, and a random satisfying program is returned:

$$\text{Sample } \rho_k \sim p_{\text{LM}}(\cdot|\texttt{prompt}(X, Y)), \text{ for } k \text{ from } 1..K \tag{2}$$

$$\text{Pick a } \rho \text{ from } \{\rho_k \ : \ \rho_k(X_i) = \rho_k(Y_i), \forall i\} \tag{3}$$

**Fine-tuning** improves the above approach in a conceptually straightforward way. Given a dataset comprising tuples of programs and I/O examples, $\{(\rho, X, Y)\}$, we fine-tune the LM to predict a program from its input-outputs. But this dataset is hard to come by: Although there are web-scale corpora of naturally occurring source code, there is no analogous dataset of runnable code snippets paired with representative input-output examples, and this data deficit is especially true for new or unusual applications of PBE, such as the graphics programs we consider.

To assemble a large dataset of $(\rho, X, Y)$ triples we start with a small manually-constructed seed dataset, $\mathcal{D}_{\text{seed}}$, and then randomly generate new programs $\rho$ and inputs $X$ by prompting an LLM with members of $\mathcal{D}_{\text{seed}}$. The output $Y$ comes from running $\rho$ on $X$. The seed dataset effectively defines a prior over $(\rho, X)$, notated $\mathcal{G}$ in Fig. 2. We sample from $\mathcal{G}$ to collect many program-input pairs, but use program execution to predict $Y$, not an LLM. The resulting dataset, which we call $\mathcal{D}_{\text{tune}}$, is used to train an LLM to generate programs when prompted with input-outputs. As this fine-tuned LLM effectively learns to do probabilistic inference in the graphical model shown in Fig. 2 (right), we write this fine-tuned LLM as $q_\theta(\rho|X, Y)$. This inference network is trained to maximize

$$\max_\theta \log q_\theta(\rho|X, Y), \text{ where } (\rho, X) \sim \mathcal{G}(\mathcal{D}_{\text{seed}}) \text{ and } Y = \rho(X) \tag{4}$$

This method is closely related to self-instruct [32] and wake-sleep [33]. Like self-instruct, we use prompting to bootstrap a large dataset from a small manually-constructed one. Our method differs by using the LLM to generate a hidden latent variable (the program) while a different generative process produces an observed variable (the program outputs). Like wake-sleep, we use samples from a generative model to train an inference network, but we do not further train the generative model itself. Next, we will see that bringing the method much closer to wake-sleep by updating the generative model plays an important role when deploying the system on out-of-distribution problems.

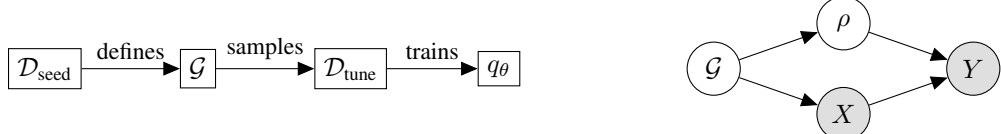

Figure 2: Left: Data generation pipeline. Right: The fine-tuned network $q_\theta$ learns to do inference in a graphical model where the prior over programs, $\mathcal{G}$, is defined by prompting an LLM with example code in $\mathcal{D}_{\text{seed}}$, while the likelihood $p(Y|\rho, X)$ is defined by program execution.

**Adaptation.**   One of the most powerful features of source code as a representation is its ability to efficiently express a wide range of computations. Therefore it is of interest to study the ability of fine-tuned LLMs to extrapolate to PBE problems outside the distribution of the fine-tuning data.

We consider a basic approach to adapting to a different distribution of problems, assuming access to problems drawn from the testing distribution, but without labeled program solutions. This mimics the deployment of PBE systems to end-users who may have their own idiosyncratic distribution of problems they care about, and who do not provide ground-truth programs, but who can provide feedback on if a generated program has correct behavior. This means we have an unlabeled dataset $\mathcal{D}_{\text{adapt}}$ comprising input-outputs $(X, Y)$, as well as a labeled seed dataset $\mathcal{D}_{\text{seed}}$ comprising triples $(\rho, X, Y)$. Adaptation proceeds by iterating between pretraining with $\mathcal{G}(\mathcal{D}_{\text{seed}})$, testing on $\mathcal{D}_{\text{adapt}}$, and adding back into $\mathcal{D}_{\text{seed}}$ any program solutions found on the adaptation problems, which then become seeds for the next iteration. This produces a sequence of fine-tuned models, indexed below by $i$:

$$\text{train model:} \quad \theta^i = \arg\max_\theta \log q_\theta(\rho|X,Y), \text{ where } (\rho, X) \sim \mathcal{G}(\mathcal{D}_{\text{seed}}^i) \text{ and } Y = \rho(X)$$

$$\text{run inference:} \quad \rho_k^{X,Y} \sim q_{\theta^i}(\rho|X,Y) \text{ for } (X,Y) \in \mathcal{D}_{\text{adapt}} \text{ and } k \text{ from } 1..K$$

$$\text{update seed:} \quad \mathcal{D}^{i+1} = \mathcal{D}^i \cup \left\{ (\rho_k^{X,Y}, X, Y) \; : \; (X,Y) \in \mathcal{D}_{\text{adapt}}, \; k \in [K] \text{ if } \rho_k^{X,Y}(X) = Y \right\} \tag{5}$$

The equations can be seen as a wake-sleep algorithm where "dreaming" corresponds to training $q$ on fantasy data (first equation) while "waking" corresponds to running inference and updating the generative model $\mathcal{G}$ (by updating the seed, second pair of equations). Ideally, each cycle of this wake-sleep adaptation solves more out-of-distribution problems, which tugs the generative model $\mathcal{G}$ toward the target distribution, unlocking solutions to more out-of-distribution problems, etc. This hinges on each iteration actually solving new problems from the unlabeled dataset. Theoretically this is guaranteed given enough inference-time compute (large $K$ above). We explore in Sec. 4.3 the extent to which this holds in practice.

## 4   Experiments

We study different LLM-approaches to programming-by-examples across three domains (Fig. 1):

1. **List functions** is a PBE domain meant to model a "programmer's assistant". It concerns discovering algorithms that transform lists of numbers, given input-output examples. This problem statement has a long history within program synthesis [13, 34], and was popularized within machine learning by DeepCoder [35]. We consider two modern list function datasets created by Rule et al. 2024 [17] and Shi et al. 2023 [12], which both involve higher-order functions and nontrivial procedures such as map, filter, and sort. Rule et al. was recently added to BigBench [36].

2. **Text editing** is a domain where a program synthesizer assists an end-user edit their spreadsheets or other documents. From string-to-string examples, the system generates edit macros for tasks such as reformatting dates, extracting fields from semistructured text, etc. [2, 37, 38, 4]. Text editing is the most prominent commercial success of PBE: The FlashFill PBE system ships in Microsoft Excel and is used by many millions of people [6]. We consider two text editing datasets: SyGuS problems [22]—which are easier—and PROSE [39] problems, which constitute the most challenging dataset of its kind [38].

3. **LOGO/Turtle graphics** is a domain whose goal is to synthesize a program that generates a target image.[1] Systems of this kind can be used both for high-level visual reasoning and for helping artists make structured edits to images [40, 41]. We use a dataset of geometric designs expressed as LOGO/Turtle [42] programs—where the programs move a simulated pen over a canvas—taken from Wong et al. [43]. To allow the LLM to visually perceive the input image, we convert the image to ASCII-art style strings; see Fig. 5 and Appendix. A.1.3.

---

[1]This is PBE with a single example and null input, effectively compressing the image into a program.

## 4.1 How well does the fine-tuned model perform?

We prepare seed datasets for each domain, synthetically generate a large training set, and then fine-tune a DeepSeekCoder LLM [44] that was pretrained on source code.[2] For list functions we seed with 50 problems from Rule et al. 2024; For text editing, we consider seeding with either SyGuS or a 40-problem subset of PROSE; for LOGO we seed with 200 training-set problems in Wong et al. [43].

The resulting fine-tuned models are surprisingly effective within their respective PBE domains. On list functions our finetuned model surpasses the best symbolic search baselines reported in Rule et al. (Fig. 3a), surpasses the best neurosymbolic search method from Shi et al. (Appendix Fig. 10), and surpasses GPT4. It also solves 100% of the list to list benchmark problems from $\lambda^2$ (a well-known symbolic synthesizer), shown in Appendix Tbl. 4: although plausibly, many $\lambda^2$ problems are in the pretraining data. On text editing, it surpasses the performance of FlashFill and approaches the level of FlashFill++ (Tbl. 1, Fig. 3b). On LOGO, it solves 90% of the test set (Fig. 3c), surpassing systems such as DreamCoder [45], which introduced the first version of these LOGO problems. It also solves more problems than LILO and Regal [43, 46], which are LOGO program synthesizers that input natural language describing how the image should be drawn. In contrast, our model does not use any language clues, generating purely from the image. In addition to quantitatively solving more problems, we note that there are qualitative improvements to the breadth of problems that can be solved in the first place because the LLM can generate Turing-complete code spanning a much broader space of computations (Fig. 4).

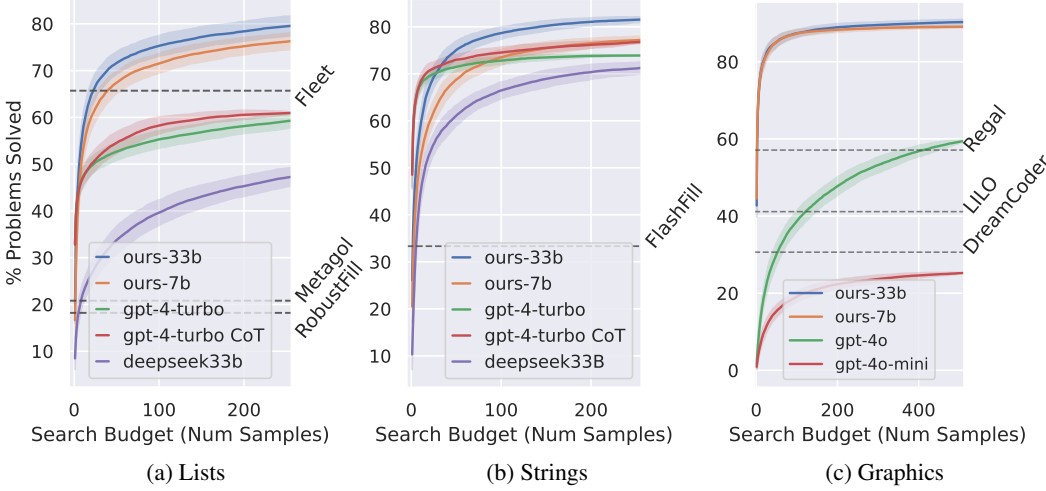

Figure 3: Test set performance. A problem is solved if the predicted program generates correct outputs on the holdout inputs. Metagol [47], RobustFill [20], and Fleet [48] results taken from [17]

There are caveats to the above results. First, the fine-tuned model essentially never produces a correct program on the first try: It requires tens or hundreds of samples, each of which is compared against the ground-truth input-outputs, and discarded if it contradicts the examples. On a GPU like the one we use (an Nvidia A6000) this rejection sampling takes on the order of a few minutes to solve a given problem. However, compared to classic enumerative program synthesizers [13, 49], or even compared to those with neural guidance [35, 22], proposing a few thousand programs is relatively little, and could not plausibly cover a significant fraction of the exponentially large search space.

|  | gen. accuracy | oracle accuracy |
|---|---|---|
| FlashFill | 33% | — |
| FlashFill++ | — | ≈100% |
| ours, 33B | 82% | 88% |

Table 1: Generalization accuracy: % problems where the program makes correct predictions on every holdout test. Oracle accuracy: % problems where a correct program was generated (even if incorrect programs were also generated that also passed the training input-outputs). Flash-Fill++ [38] only reports oracle accuracy. [3]

---

[2]We prefer DeepSeek because it is roughly LLaMA-level, but has fully open training details.
[3]The FlashFill results were obtained using Microsoft Excel for Mac.

| INPUT | OUTPUT |
|---|---|
| Mary had a little lamb
Its fleece was white... | 1:Mary had a little lamb
2:Its fleece was white... |
| Twinkle, twinkle, ...
How I wonder what you...
Up above the world so...
Like a diamond in the... | 1:Twinkle, twinkle, ...
2:How I wonder what you...
3:Up above the world so...
4:Like a diamond in the... |

```
generated program
lines = text.splitlines()
result = ""
for i, line in enumerate(lines):
    result += f"{i + 1}:{line}\n"

return result[:-1]
```

| INPUT | OUTPUT |
|---|---|
| NY | New York |
| CA | California |
| AK | Alaska |

```
generated program
states = {
    "AL": "Alabama",
    "AK": "Alaska",
    .........
}
return states[text]
```

Figure 4: PBE with LLMs allows using general-purpose programming languages which can mix string and numerical operations in ways not allowed by domain-specific languages [38] (top), and allows world knowledge to inform code generation (bottom). I/Os and code partly elided for space.

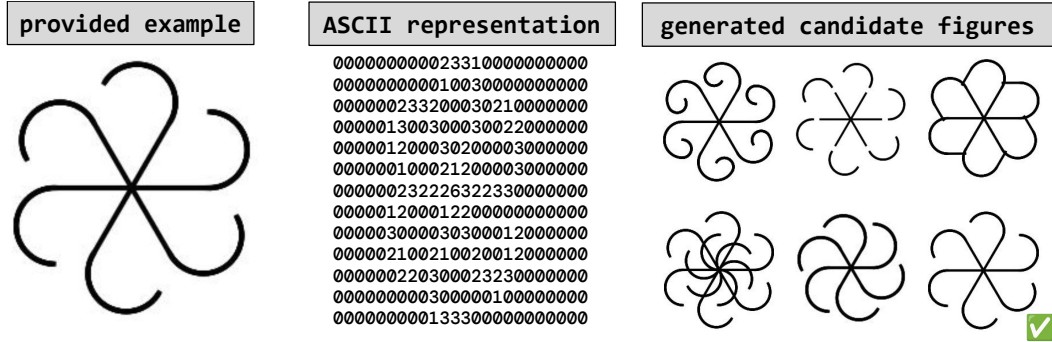

Figure 5: ASCII representation of LOGO graphics. Average pixel intensity indicated by numbers 0-9

The second caveat is that the model degrades when tested out-of-distribution. An example of this degradation is illustrated in Fig. 6, which tests the LOGO graphics model on hand drawings (after training on clean computer graphics). On the out-of-distribution hand drawing the model mostly samples programs that do not fit the data, but its accuracy does not fall to zero, meaning that with enough compute budget, it does actually generate reasonable programs. This foreshadows the results in Sec. 4.3, which more systematically studies out-of-distribution behavior.

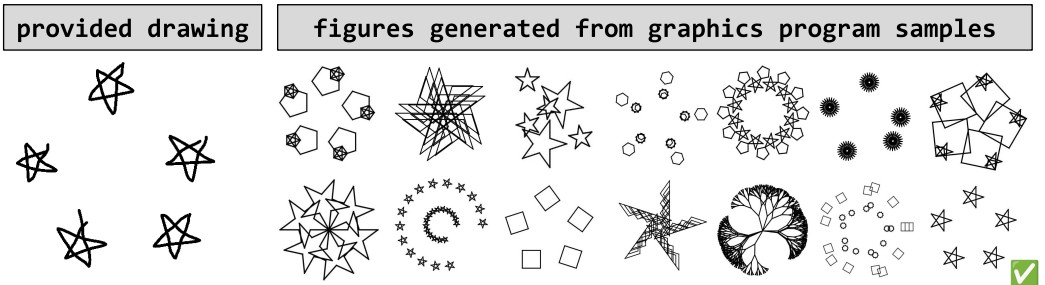

Figure 6: Example out-of-distribution LOGO test: inferring a graphics program from a hand drawing. See also Appendix Fig. 12

## 4.2 What causes the fine-tuned model to succeed or fail?

Classic symbolic approaches to PBE, when they are based on enumeration, tend to succeed whenever the target program is syntactically small. Approaches based on clever dynamic programming, such as the FlashFill family [4], succeed when the program is representable in the domain-specific language. What predicts success for these LLM approaches?

To answer this question we investigate several hypotheses. First, potentially the success is determined by program size, and degrades as programs grow longer. Second, as a more refined notion of size, we instead measure the description length *under the prior*, which for a program $\rho$, is $-\log p_{\text{LM}}(\rho|\mathcal{G}(\mathcal{D}_{\text{seed}}))$. Description length under the prior would be a good predictor of success if the fine-tuned model engages in blind guess-and-check: simply learning the distribution $\mathcal{G}(\mathcal{D}_{\text{seed}})$, and sampling from this prior while ignoring the input-outputs. Third, one possibility is that success is predicted by description length *under the approximate posterior* $(-\log q_\theta(\rho|X,Y))$, which would be the case if the fine-tuned model attends closely to the input-outputs and reshapes its distribution accordingly, instead of defaulting to the prior. To test these hypotheses we calculate the average compute budget needed to solve each problem, and compare it with these different variables. Fig. 7 shows that posterior description length is more predictive than program size and prior description length: unlike classical methods, metrics of program length correlate poorly with problem difficulty, and there is no evidence that the fine-tuned model's behavior can be characterized as blind guess-and-check. (See also Fig. 5).

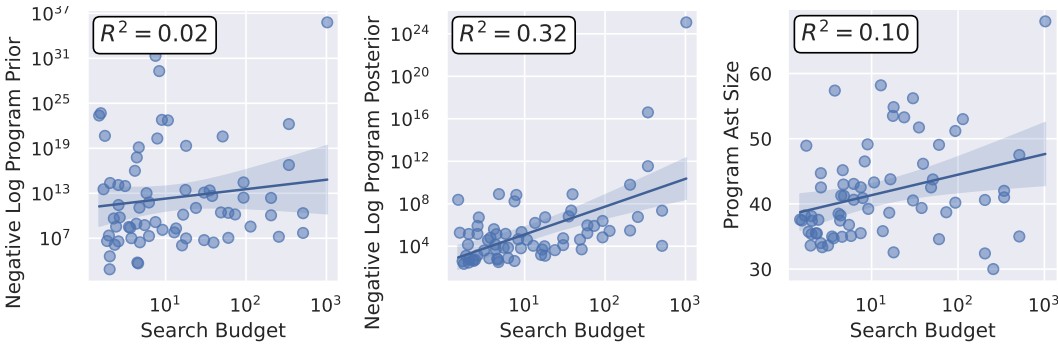

Figure 7: Compute budget needed to solve a problem is best predicted by description length under the approximate posterior, *not* program size or prior description length, suggesting that the fine-tuned model is not engaging in blind guess-and-check.

## 4.3 Out-of-distribution generalization

One advantage of classic symbolic PBE methods is that they do not make statistical assumptions about their test problems. Indeed, some classic methods can, within their domains, synthesize programs perfectly (i.e. always find a program that fits the training input-outputs). In contrast, neural networks can struggle to generalize beyond the training distribution.

We therefore consider train/test splits that force the model to generalize beyond the distribution of its training data (beyond $\mathcal{D}_{\text{seed}}$). On text editing, we seed with SyGuS problems, and perform out-of-distribution testing on PROSE problems (PROSE is much harder than SyGuS). On list functions, we seed with problems from Rule et al. 2024 and test on Shi et al. 2023 (the Shi dataset contains unusual combinators, such as `Scan`). On LOGO, we seed with short programs ($\leq 12$ lines of code), and test on long programs ($> 12$ lines of code). Using these splits we also measure the ability of the adaptation method in Sec. 3 to improve out-of-distribution generalization.[4]

Fig. 8 shows that there is nontrivial degradation when testing out of distribution. For example, a 7B model seeded with PROSE problems and tested on a different subset of PROSE has an accuracy of 76% (Fig. 3b), but this degrades to 59% when seeded with SyGuS problems, which follow a different distribution and are generally simpler and easier than PROSE (Fig. 8b).

---

[4]We work here with 7B models because Sec. 4.1 found that fine-tuned 33B models are only slightly better than 7B, and 7B is cheaper to run.

We further perform the adaptation method described in Sec. 3 in order to measure the extent to which it can narrow these domain gaps. In every case it allows solving more out-of-distribution problems, increasing absolute performance by around 10% or more in all domains, which is a relative increase of about 16% for text/list and a relative increase of about 190% for LOGO (approximately tripling the number of solved LOGO problems).

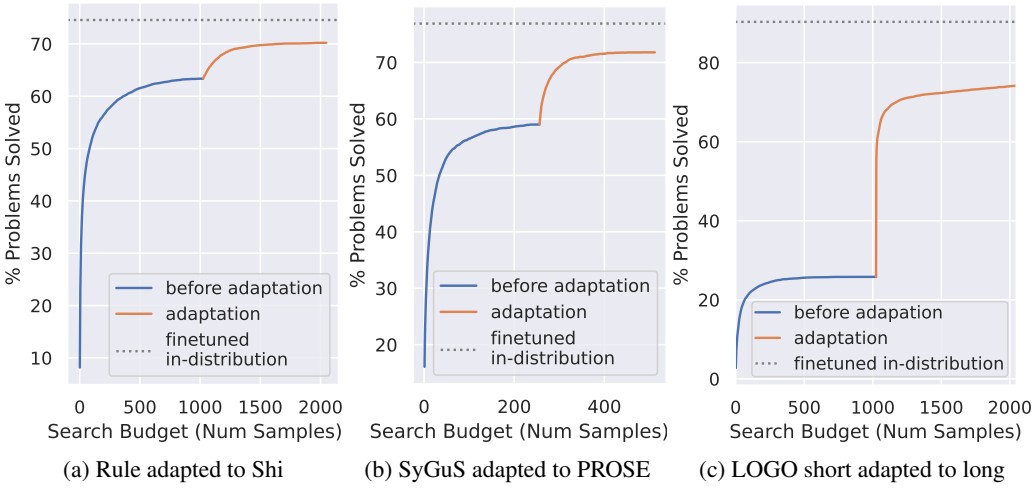

(a) Rule adapted to Shi     (b) SyGuS adapted to PROSE     (c) LOGO short adapted to long

Figure 8: Out-of-distribution generalization and adaptation to new test distribution.

To better understand the dynamics of adaptation, we visualize the specific problems solved before and after adaptation on LOGO graphics (Fig. 9). Before adaptation, only a handful of out-of-distribution problems are solvable, and only with a significant search budget. Adaptation allows the system to quickly solve similar out-of-distribution problems in the future, but does not allow the system to generalize to problems very unlike those originally solvable by the fine-tuned model. In principle, expanding the inference-time compute budget should allow successful adaptation (large $K$ in Eq. 5). Another more compute-efficient approach would be to increase the amount of adaptation data by introducing 'steppingstone' problems in the adaptation set that give a gentler transition from the original training distribution.

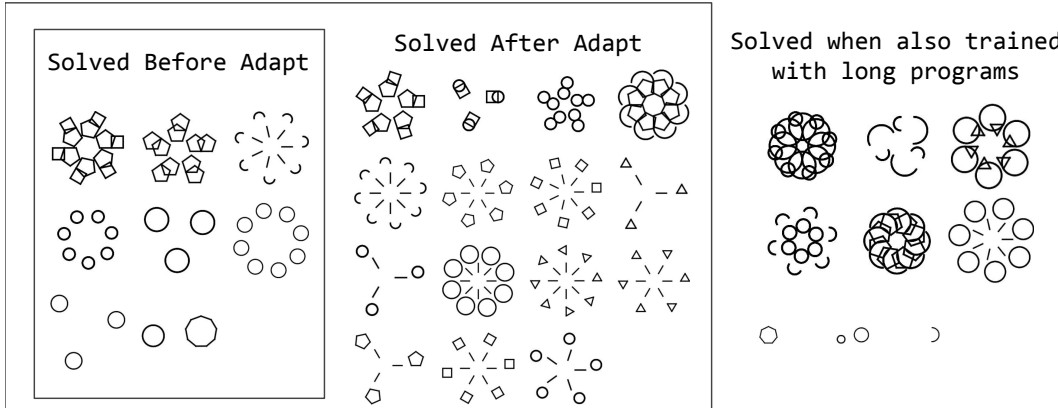

Figure 9: Out-of-distribution LOGO problems (requiring long programs with $> 12$ lines of code). We show example problems that are solved by the original model fine-tuned on short programs, which then become training data for the next round of adaptation. Adaptation allows consistent solving of problems similar to those that the original fine-tuned model could sometimes solve, but is not a panacea: Problems dissimilar to those solved by the initial model are not ever correctly generated, despite the fact that they are solvable by a model fine-tuned in-distribution.

# 5 Related Work

**Automatic data generation with LLMs**, such as self-instruct [32], WizardCoder [50], and many others [51, 52, 53, i.a.], works by prompting an LLM to produce outputs which are then used for later learning stages such as fine-tuning. These approaches are applied recursively to their own output: Previously generated data is incorporated into future prompts. We similarly generate a dataset $\mathcal{D}_{\text{tune}}$ by prompting an LLM with $\mathcal{D}_{\text{seed}}$, but (1) do not recursively prompt the LLM with its own outputs and (2) combine the LLM generations with program execution to make program outputs. This gives a different mathematical interpretation to our data generator. First, the programs are samples from a prior, $\mathcal{G}$, defined by $\mathcal{D}_{\text{seed}}$, which would not be a valid interpretation if the LLM was repeatedly fed its own outputs. Second, there is an observation model or likelihood function, $p(Y|\rho, X)$, which is defined not by the LLM, but by a Python interpreter. In this way, our data generator constructs training examples for fine-tuning that teach the network how to invert the execution process of the Python interpreter.

**Machine learning applied to PBE** has often sought to accelerate search: to find *any program at all* consistent with the I/O examples [19, 20, 45, 21, 35, 22, 12, 54, 55, 56], which is nontrivial due to the combinatorial nature of the search, even after confining to a domain-specific programming language. A complementary line of research explores inductive biases that favor programs likely to generalize to new inputs, such as learning a prior or ranking function [57, 41, 58, 59]. Our work should be seen within the tradition of learning to search for programs. We show that finetuned models serve as an effective yet simple foundation for accelerating search in PBE, allowing search to be tractable over much richer and more expressive languages such as Python.

**Classic PBE.** Traditional approaches to programming-by-examples operate by symbolically searching or solving for programs consistent with the input-output examples [13, 49, 2, 1, 37, 6]. They use domain-specific programming languages that are designed to either enable efficient search and/or bias the system toward functions that are likely to generalize new inputs. Search for programs can even be polynomial time when this domain-specific language has a special structure (roughly, when every function can be 'inverted'), a key enabler of FlashFill, the first commercial success of PBE [4, 2].

**LLMs as inductive reasoners.** Using an LLM to perform inductive reasoning—to generate abstract hypotheses from concrete specific examples—has been explored by several recent works [29, 30, 60, 61], all of which has found significant value in translating these hypotheses into programs, and all of which have worked by prompting pretrained GPT-style models. Our work can be seen as helping answer a natural question posed by these previous works: Given that LLMs can generate hypotheses from examples, can they produce programs of the nature and complexity demanded by PBE? We find this is largely the case after fine-tuning, both for classic PBE domains and unusual ones.

**Self-Debugging, Refinement, and Self-repair.** One way of improving the code generation abilities of an LLM is to have it attempt to debug its own code whenever the initially generated code does not pass the provided test cases [62, 63, 64, 65, 66, 24, 67]. We did not explore this strategy, however, because a more basic approach that simply regenerated a new program from scratch already surpassed the prior state of the art (both symbolic and neural baselines), provided we finetune. However, further pushing the boundary of PBE may benefit from self-debugging strategies.

**Ranking LLM-generated code.** Past work considers a variety of ways to select an output from a collection of LLM-sampled programs [23, 59, 68, 11], many of which are more sophisticated than simply filtering by the examples, which is what we do here. Like with self-debugging, integrating these techniques should be synergistic with our approach.

# 6 Limitations

Our work has important limitations. From an engineering perspective, using a 7B-33B neural network to perform PBE is not practical for most end-users, who may be doing PBE on their laptop or desktops in order to accomplish small one-off tasks. For this reason, true deployment to end-users may require

investigating the effectiveness of much smaller neural networks (not an LLM), and it may also be valuable to study the effect of network compression and distillation upon our finetuned models.

From the perspective of understanding where and why our system succeeds and fails, we have shown that neither program size nor likelihood under the prior suffice to predict success, finding the posterior likelihood is a better predictor, albeit an imperfect one. Although this allows us to discard the hypothesis that the system is merely sampling from the prior, it also just pushes the question back one stage further: What exactly about specific problems causes the neural network's approximate posterior to put more or less probability mass on correct solutions? While in classic PBE one can obtain sharp answers as to why a certain problem was solved or not, this is a much harder question with neural networks, whose workings are more opaque.

## 7  Discussion

PBE with fine-tuned LLMs is surprisingly effective, surpassing many of the best neural and symbolic baselines we know of, even for uncommon domains such as LOGO graphics. Why is that? Fundamentally, the neural network only needs to act as a heuristic proposer of solutions, because we can check against the input-outputs. Therefore, one possible explanation is that the tendency of language models to over-generate, hallucinate, and cover the long tail of possibilities is actually an asset, instead of a liability. And although there is a degree of degradation on out-of-sample problems, the degradation is not so severe that out-of-distribution problems become utterly unsolvable: Instead, they merely become harder to solve, a phenomenon that allows adaptation to work in the first place.

Simultaneously one should be hesitant about claiming that PBE is 'solved.' Optimistically, current PBE benchmarks exist to test the frontier of what is possible, and so doing well on those benchmarks might just mean that the frontier has moved. More realistically, determining if an AI system truly works in the wild requires more than just pushing benchmark numbers, which can be misleading when those benchmarks do not capture the long tail of naturally-occurring tests. Furthermore, all AI systems present tradeoffs, and a neural system's unpredictability, high computational cost, and out-of-distribution fragility should be weighed against whatever high benchmark numbers they may achieve. Despite these caveats, we are optimistic about the promise of tuning LLMs for PBE, and believe that it has the potential to dramatically expand the scope of solvable problems and even solvable domains.

**Acknowledgements.**   We are grateful for assistance from Joshua Rule in the processing of the list functions data, and for feedback from Yewen Pu on the manuscript. This work was supported by an NSF CAREER grant as well as gifts from Google and Cisco.

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

# A    Appendix / supplemental material

## A.1    Experiment Details

We used a temperature of 1.0 for sampling in our experiments unless otherwise stated. All experiments were performed on single-node machines (8xA6000 or 8xA100, etc.) without a multi-node distributed computing setup.

### A.1.1    List Tasks

We selected 50 problems from Rule et al. as our seed set, reserving the remaining problems for testing. To ensure comparability with Rule et al., we tested on the first 100 problems (excluding those in the seed set), resulting in 77 test problems. We consistently used 10 input-output examples, with the remaining 54 examples serving as a held-out test set. When filtering duplicate synthetic data, we employed an open code embedding model[69] available on Hugging Face. As a sanity check, we also use 10 list to list problems from $\lambda^2$ benchmark and shown the model can effectively solve them in Table.4

### A.1.2    String Tasks

We utilized 100 string-to-string/null transformation problems from the prose-benchmark. When available, we used 10 input-output examples, always reserving at least one example as a hold-out test set. This ensures the generalization of our synthesized programs, as the benchmark did not provide held-out data.

For the FlashFill baseline, we used Microsoft Excel for Mac version 16.8. We opened each individual xlsx file containing the test problem examples and manually triggered the FlashFill function by pressing Ctrl+E.

### A.1.3    Logo Tasks

To facilitate graphics input inference with code language models, we converted logo graphics into ASCII-represented strings, as shown in Figure 5. For each input image, we cropped a 512x512 section from the center and then divided it into 32x32 blocks, each with a size of 16x16. We counted the number of black pixels in each block, calculated their density ($\frac{\text{num black pixels}}{16 \cdot 16}$), and quantized this density value into 10 levels, represented by the ASCII numbers 0-9. By representing each block with an ASCII number, an input image is represented with a string of 32 lines, and each line has 32 numbers.

For the turtle graphics program, we adopted the Python turtle graphics program from Regal[46] with a minor modification of changing the 'embed' function to use a 'with' context manager instead, calling it 'fork_state'. This allows for equivalent but more readable code.

For the GPT-4o and GPT-4o mini multimodal baselines, we directly use images as inputs. (See Section A.5.8 for the prompt template.)

## A.2    Syntheic Dataset Generation and Training Parameters

We present the dataset generation and training parameters in Table. 2 and Table. 3.

## A.3    Adaptation Implementation Details

For adaptation experiments, we generally followed the settings described above, with a few specific differences detailed below.

### A.3.1    String Tasks

To induce a domain gap (easier problems in Sygus compared to the harder, noisier problems in the Prose Benchmark), we first fine-tuned a model using Sygus problems and then tested it on the Prose Benchmark. Due to the noisy nature of the Prose Benchmark (some problems have very few examples), we adopted a setting where we utilized all the test cases to select which problems were

|  | **List** | **String** |
|---|---|---|
| **Seed Dataset Source** | Rule et al. | Prose (FlashFill++) Problems |
| **Seed Dataset Size** | 50 | 40 |
| **Synthetic Data Generator** | deepseekcoder-33b-instruct | deepseekcoder-33b-instruct |
| **Synthetic Dataset Size** | 10k | 10k |
| **Sampling Tempereature** | 0.8 | 1.0 |
| **Similarity Filter** | code embedding model | - |
| **Filter Ratio** | around 1/3 (threshold=0.93) | - |
| **Synthetic Data Prompt** | 4-shot examples | 10-shot examples |
| **LoRA Finetuning** | | |
| **Model Used** | deepseekcoder-1.5-7b-instruct | deepseekcoder-1.5-7b-instruct |
| **LoRA Rank** | 1024 | 256 |
| **LoRA $\alpha$** | 1024 | 256 |
| **Learning Rate** | 2.00E-04 | 2.00E-04 |
| **LR Schedule** | cosine | cosine |
| **Warmup Steps** | 10 | 10 |
| **Epoch** | 1 | 1 |
| **Batchsize** | 32 | 16 |
| **LoRA** | | |
| **33b Model Used for FT** | deepseekcoder-33b-instruct | deepseekcoder-33b-instruct |
| **LoRA** | 256 | 128 |
| **LoRA $\alpha$** | 256 | 128 |
| **Learning Rate** | 2.00E-04 | 2.00E-04 |
| **LR Schedule** | cosine | cosine |
| **Warmup Steps** | 10 | 10 |
| **Epoch** | 1 | 1 |
| **Batchsize** | 32 | 32 |

Table 2: List task and String task synthetic dataset generation and finetuning parameters.

solved and then used them as the seed programs for adaptation. This resulted in 64 solved problems out of the 100 problems in the benchmark.

### A.3.2   List Tasks

To obtain the finetuned in-distribution result in Fig. 8a, we fine-tuned on a synthetic dataset generated by seeding with 20 out of 100 problems from LambdaBeam, and tested on the remaining 80 problems.

### A.3.3   LOGO Tasks

For LOGO adaptation experiments, we induce domain gap by using the shorter programs (LoC $\leq 12$) of the training set, and tested on the longer programs (LoC $> 12$). The shorter programs training seed consists of around 80% problems (156 out of 200) from the original training set. The test set consists of 31 problems out of 111 problems from the original test set.

### A.4   Model Performance on LambdaBeam Benchmark

We present the results of both our 7B and 33B models on the LambdaBeam benchmark in Figure 10. We observed that even without fine-tuning for this specific benchmark, and instead fine-tuned for the list-to-list problems from Rule et al., our models performed exceptionally well, surpassing the state-of-the-art results specifically designed for the LambdaBeam problems [12].

### A.5   Prompts Used in the Experiments

### A.5.1   Syntheic Data Generation Prompt

List

|                          | **Logo**                      |
| ------------------------ | ----------------------------- |
| **Seed Dataset Source**  | Regal Python Logo Programs    |
| **Seed Dataset Size**    | 200                           |
| **Synthetic Data Generator** | deepseekcoder-33b-instruct |
| **Synthetic Dataset Size** | 32k                         |
| **Similarity Filter**    | -                             |
| **Filter Threshold**     | -                             |
| **Synthetic Data Prompt** | 6-shot examples              |
| **LoRA Finetuning**      |                               |
| **Model Used**           | deepseekcoder-1.5-7b-instruct |
| **LoRA Rank**            | 512                           |
| **LoRA $\alpha$**        | 512                           |
| **Learning Rate**        | 2.00E-04                      |
| **LR Schedule**          | cosine                        |
| **Warmup Steps**         | 20                            |
| **Epoch**                | 3                             |
| **Batchsize**            | 64                            |
| **LoRA Finetuning**      |                               |
| **Model Used**           | deepseekcoder-33b-instruct    |
| **LoRA Rank**            | 512                           |
| **LoRA $\alpha$**        | 512                           |
| **Learning Rate**        | 2.00E-04                      |
| **LR Schedule**          | cosine                        |
| **Warmup Steps**         | 50                            |
| **Epoch**                | 3                             |
| **Batchsize**            | 64                            |

Table 3: Logo task synthetic datasets generation and finetuning parameters

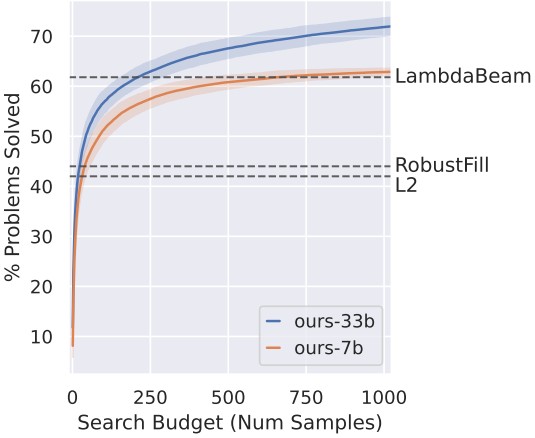

Figure 10: Performance on LambdaBeam problems from Shi et al. 2023

```
You are a CS professor. You are providing a set of challenging
    and diverse integer list to integer list function puzzle
    for your student to solve.

Puzzle 1:
Python function: input a list of integers and return a list of
    integers
'''python
{PROGRAM EXAMPLE 1}
'''
Test cases:
...
Puzzle 2:
Python function: input a list of integers and return a list of
    integers
'''python
{PROGRAM EXAMPLE 2}
'''
Test cases:
...

Following the above format, please provide 3 functions each
    follow by 10 random test cases to check the function's
    correctness full coverage
```

String

```
Excel just introduce a new feature that allows user to use
    Python to perform data transformation.
Please generate a csv file with two columns. The first column
    contains the input data and the second column contains the
    output data.
Following a accompanying python function which showcasing the
    transformation of the input data to the output data.

Here are 10 challenging examples showcaing the features

Example 1
'''csv
{INPUT OUTPUT EXAMPLE 1}
'''
Here is the Python function that help transform the first
    column to the second column.
'''python
{PYTHON EXAMPLE 1}
'''

Example 2
'''csv
{INPUT OUTPUT EXAMPLE 2}
'''
Here is the Python function that help transform the first
    column to the second column.
'''python
{PYTHON EXAMPLE 2}
'''

...
```

```
Following the above format, please provide a CSV file with two
    columns, containing between 5 to 10 rows of data showing a
    transformation from the first column to the second column.
    This csv data should illustrate a challenging and complex
    example, similar to the above examples. Following that,
    create a Python function designed to process this data. Be
    aware that this function will be tailored to not only
    accommodate the current data but also any future data that
    follows the same format or structure.
```

Logo

```
Your task is to draw simple black and white graphics with the
    custom library. DO NOT USE THE BUILT-IN TURTLE LIBRARY.
You will use a custom turtle library, similar to the built-in
    turtle library, which is sufficient for all tasks.

Here are all the available functions in the custom turtle
    library:
- forward(x): move forward x pixels
- left(theta): rotate left by theta degrees
- right(theta): rotate right by theta degrees
- penup(): stop drawing
- pendown(): start drawing
- teleport(x, y, theta): move to position (x, y) with angle
    theta
- heading(): get the current angle of the turtle
- isdown(): check if the pen is down
- forward(x): Move forward x pixels.
- left(theta): Rotate left by theta degrees.
- right(theta): Rotate right by theta degrees.
- penup(): Stop drawing.
- pendown(): Start drawing.
- teleport(x, y, theta): Move to position (x, y) with angle
    theta.
- heading(): Get the current angle of the turtle.
- isdown(): Check if the pen is down.
- with fork_state(): A context manager that runs the code in
    the block using the current context and restores the
    original state afterwards. Allows you to nest programs.
    Internally, fork_state saves the turtle state (is_down, x,
    y, heading), executes the block, then restores the original
     state.
Graphic 1
Python program: draw an interesting graphic using our own
    custom turtle library
# the following program draws ...
{PROGRAM EXAMPLE 1}
Graphic 2
Python program: draw an interesting graphic using our own
    custom turtle library
# the following program draws ...
{PROGRAM EXAMPLE 2}
...

Following the above format, please provide 5 more programs
    using our custom drawing library.
```

### A.5.2 Prompt Template for Finetuning and Zero-Shot Experiments

### A.5.3 List

```
Implement the function solve_puzzle that takes a list of
    integers and returns a list of integers. The function
    should satisfy the following assertions
assert solve_puzzle(...) == ...
assert solve_puzzle(...) == ...
assert solve_puzzle(...) == ...
...
```

### A.5.4 List (Chain-of-Thought)

```
Implement the function solve_puzzle that takes a list of
    integers and returns a list of integers. The function
    should satisfy the following assertions:
assert solve_puzzle(...) == ...
assert solve_puzzle(...) == ...
assert solve_puzzle(...) == ...
...
Please observe the relation between input and output and think
    step by step. Output the function in a markdown format in
    the end.
Solution:
```

### A.5.5 String

```
 Implement the function edit_text that takes a string and
    returns a string. The function transforms the input string
     to the output string. The function should satisfy the
    following assertions:
assert edit_text(...) == ...
assert edit_text(...) == ...
assert edit_text(...) == ...
...
```

### A.5.6 String (Chain-of-Thought)

```
Please implement the function edit_text that takes a string
    input and returns a modified string.
Note that you can import re, datetime, or any built-in Python
    library to solve the problem.
The function should satisfy the following test cases:
assert edit_text(...) == ...
assert edit_text(...) == ...
assert edit_text(...) == ...
...
Please reason through the problem and think step by step, and
    finally implement the function and output the full function
     implementation in a markdown code block in the end.
```

### A.5.7 Logo

```
Here is a gray scale images representing with integer values
    0-9.
{CONVERTED IMAGE STRING}...
```

```
...
Please write a Python program that generates the image using
    our own custom turtle module
```

### A.5.8 Logo (Multimodal Few-shot)

```
Given the following custom Turtle graphics -like library , Please
    use it to write a program that draws the given image .
'''python
from myturtle import Turtle
from myturtle import HALF_INF , INF , EPS_DIST , EPS_ANGLE
turtle = Turtle ()
def forward ( dist ):
    turtle . forward ( dist )
def left ( angle ):
    turtle . left ( angle )
def right ( angle ):
    turtle . right ( angle )
def teleport (x, y, theta ):
    turtle . teleport (x, y, theta )
def penup ():
    turtle . penup ()
def pendown ():
    turtle . pendown ()
def position ():
    return turtle .x, turtle .y
def heading ():
    return turtle . heading
def isdown ():
    return turtle . is_down
def fork_state ():
    """
    Fork the current state of the turtle .
    Usage :
    with fork_state ():
        forward (100)
        left (90)
        forward (100)
    """
    return turtle . _TurtleState ( turtle )
'''
Below are some example programs that draw the given images .
<IMAGE >
Here is a program that draws the above image .
The figure is like a medium 8 gon .
'''python
for i in range (8):
    forward (4)
    left (45.0)
'''
<IMAGE >
Here is a program that draws the above image :
The figure is like 5 sided snowflake with a medium circle and a
    medium semicircle as arms .
'''python
for j in range (5):
    with fork_state ():
        penup ()
        forward (2)
```

```
          left(0.0)
          pendown()
          for i in range(HALF_INF):
              forward(EPS_DIST*2)
              left(EPS_ANGLE)
          for i in range(HALF_INF):
              forward(EPS_DIST*2)
              left(EPS_ANGLE)
          penup()
          forward(2)
          left(0.0)
          pendown()
          for i in range(HALF_INF):
              forward(EPS_DIST*2)
              left(EPS_ANGLE)
      forward(0)
      left(72.0)"
```
```
<IMAGE>
Here is a program that draws the above image.
The figure is like 7 concentric circles.
```python
for j in range(8):
    for i in range(HALF_INF):
        forward(EPS_DIST*j)
        left(EPS_ANGLE)
    for i in range(HALF_INF):
        forward(EPS_DIST*j)
        left(EPS_ANGLE)
```
<TEST_IMAGE>
Output the program that draws the following image. Reason about
    the given image and write a program that draws it in a
    markdown code block.
```

| Name | Description | ours-7B | ours-33B |
|---|---|---|---|
| Dedup | Remove duplicate elements from a list. | ✓ | ✓ |
| Reverse | Reverse a list. | ✓ | ✓ |
| Droplast | Drop the last element in a list. | ✓ | ✓ |
| Dropmax | Drop the largest number(s) in a list. | ✓ | ✓ |
| Dupli | Duplicate each element of a list. | ✓ | ✓ |
| Evens | Remove the odd numbers from a list. | ✓ | ✓ |
| Multfirst | Replace every item in a list with the first item. | ✓ | ✓ |
| Multlast | Replace every item in a list with the last item. | ✓ | ✓ |
| Shiftl | Shift all elements in a list to the left. | ✓ | ✓ |
| Shiftr | Shift all elements in a list to the right. | ✓ | ✓ |

Table 4: 10 list $\mapsto$ list functions from $\lambda^2$ [13]

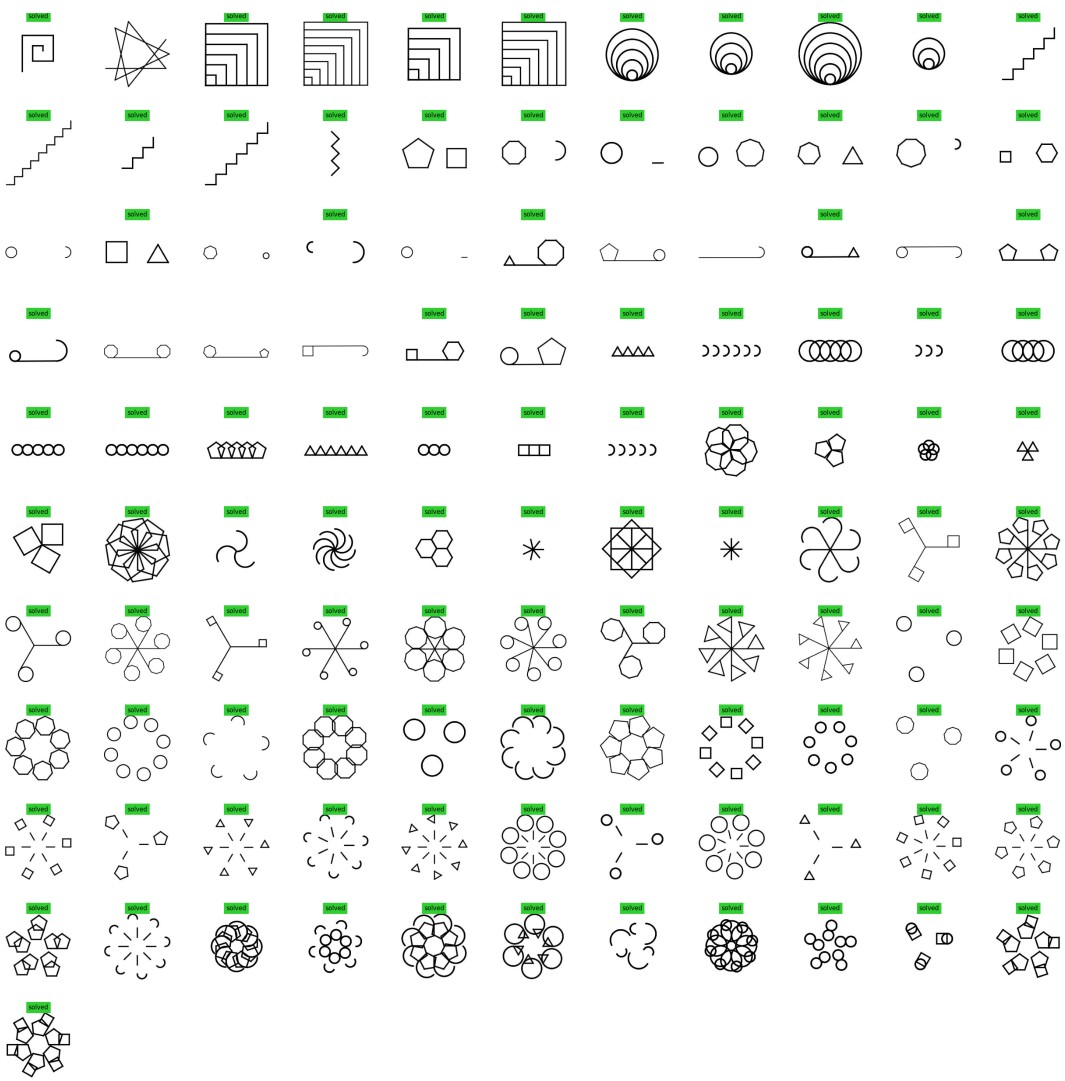

Figure 11: All LOGO test problems: problems solved by our finetuned 33b model are marked as green

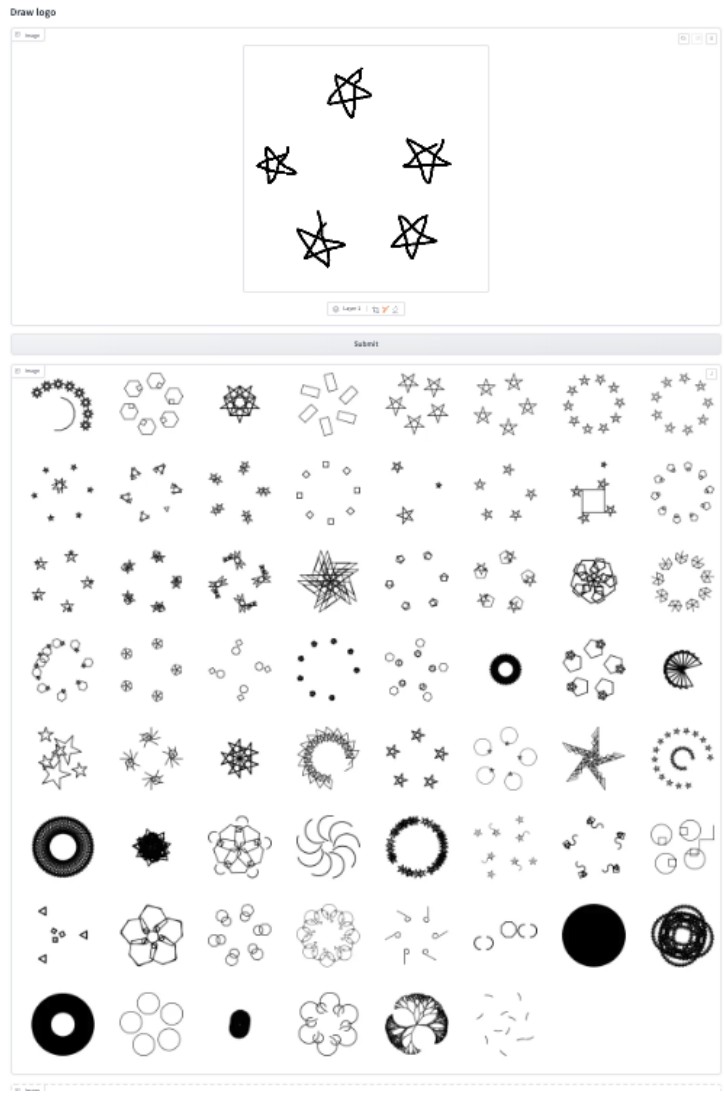

Figure 12: Hand-drawn LOGO test showing every generated sample. We built a graphical interface to allow users to draw images as input. The sample budget for this demo is 64.

