# OpenReview forum: "Is Programming by Example Solved by LLMs?"
_NeurIPS.cc/2024/Conference — NeurIPS 2024 poster_

### Official Review · Reviewer_k14y · 2024-06-28

**Soundness:** 3
**Presentation:** 3
**Contribution:** 2
**Rating:** 5
**Confidence:** 4

**Summary:**

This paper investigates the effectiveness of Large Language Models (LLMs) in solving Programming-by-Example (PBE) tasks. Evaluations are conducted on three classic PBE domains including lists and strings, as well as a graphics programming domain. The findings suggest that while pretrained LLMs are not inherently effective for PBE, fine-tuning significantly enhances their performance on in-distribution tasks.

**Strengths:**

- Thorough evaluation and detailed analysis.

- Clear cases and illustrations.

- Addressing the challenge of small datasets for fine-tuning LLMs.

**Weaknesses:**

- In the experiments, there are no LLM competitors in the graphics domain. Any reasons?

- Why are only FlashFill and LambdaBeam compared in the experiments of Figure 6?

- The adaptation method used to improve out-of-distribution performance exposes the model to the test set content beforehand. Especially in string tasks, directly selecting the adaptation seed program from all test cases may be unfair.

- The examples used in the experiments are relatively weak and do not closely resemble real-world programming tasks.

- If the adaptation's seed program is not provided, even after fine-tuning, the out-of-distribution generalization ability of LLMs still appear to be quite weak.

Typos:
in abs: potentially increasingly the flexibility -> potentially increasing the flexibility

**Questions:**

- How does GPT-4 perform on the entire PROSE dataset?

- Would the key factors that lead to the success or failure of LLMs differ across problems in three different domains?

---

> ### Author Rebuttal · Authors · 2024-08-05
>
> Thank you for the thoughtful review. Please see below for a new experimental results that you suggested we run, together with our responses to your questions.
>
> > In the experiments, there are no LLM competitors in the graphics domain. Any reasons?
>
> Thank you for your suggestion! We added the GPT-4o and GPT-4o-mini multimodal model results with image input. We summarize the results on the graphics domain as below:
>
> |System |Method Type|Accuracy|
> |---|---|---|
> |Ours-33b  | LLM + Symbolic | 90%|
> |Ours-7b                               | LLM + Symbolic       | 89% |
> |GPT-4o                        | VLM + Symbolic    | 59%|
> |Regal (ICML’24)           |LLM + Library Learning | 57%|
> |LILO (ICLR’24)             | LLM + Symbolic  | 41%|
> |DreamCoder (PLDI’21) |Neurosymbolic             | 31%  |
> |GPT-4o-mini                        | VLM + Symbolic    | 25%|
>
> (We updated our LOGO results because we found a small code issue post-submission; fixing that issue significantly improved our system, hence why our system's numbers above are better from the original submission, but the qualitative conclusions are the same.)
>
> > The examples used in the experiments are relatively weak and do not closely resemble real-world programming tasks.
>
> We disagree for the following reasons:
>
> 1. Text editing PBE is used *millions of times every week* [1]
>
> 2. PBE is not about e.g. repo-level edits, but is instead about creating individual subroutines, including for users that can't even program. Accordingly the datasets we use cover real-world situations where an individual subroutine is desired. For example, the Shi et al. List dataset comprises 100 programming tasks manually designed to be interesting and useful in practice, while PROSE/SyGuS test common spreadsheet operations.
>
> 3. Our benchmarks are typical ones for PBE on neural/symbolic systems, allowing comparisons of LLMs against the recent neurosymbolic literature, done as follows:
>
>
> |Domain| Work|               Venue|
> |---|----|----|
> |List |  LambdaBeam| NeurIPS ‘23|
> ||Fleet|Nature Comms ‘24|
> |LOGO| LILO|               ICLR ‘24|
> ||  Regal|              ICML ‘24|
> || DreamCoder | PLDI ‘21|
> |String| FlashFill++|   POPL ‘23|
>
> [1] https://blog.sigplan.org/2021/09/14/the-story-of-the-flash-fill-feature-in-excel/
>
> > Why are only FlashFill and LambdaBeam compared in the experiments of Figure 6?
>
> For lists, LambdaBeam is the primary comparison because it was designed specifically to solve the list benchmark Fig 6 evaluates on (and it does well on that benchmark).
>
> For strings, FlashFill is the primary comparison because we are evaluating on holdout test inputs, and FlashFill++ does not report numbers for holdout tests (and FlashFill++ is not publicly available so we can't run it ourselves).
>
> > The adaptation method used to improve out-of-distribution performance exposes the model to the test set content beforehand. Especially in string tasks, directly selecting the adaptation seed program from all test cases may be unfair.
>
> Motivated by your comment we ran the OOD-adapted model on a random sample of never-before-seen PROSE problems (unseen during adaptation). We see that the domain gap is approximately halved:
>
> |strings| train SyGuS, test PROSE|
> |----|----|
> |before adaptation  |57.7%|
> |after adaptation  | 65.2%|
> |finetune in-distribution | 72.5% |
>
> > If the adaptation's seed program is not provided, even after fine-tuning, the out-of-distribution generalization ability of LLMs still appear to be quite weak.
>
> Yes! Without (unlabeled) adaptation data, out-of-distribution generalization is quite weak. We hope to paint a nuanced picture of LLM abilities rather than claim LLMs are universally better along every dimension.
>
> > Would the key factors that lead to the success or failure of LLMs differ across problems in three different domains?
>
> Primarily if the problems are in-distribution, which is why we focus on OOD adaptation methods.
>
> Thank you for your input and please let us know if you have any further questions.

---

> > ### Comment · Reviewer_k14y · 2024-08-14
> >
> > Thanks for the responses, which have addressed some of my concerns. I have increased the score.

---

> > > ### Author Response · Authors · 2024-08-14
> > >
> > > Thank you for your support and for helping us improve our work!

---

### Official Review · Reviewer_mgYE · 2024-07-10

**Soundness:** 3
**Presentation:** 4
**Contribution:** 2
**Rating:** 6
**Confidence:** 5

**Summary:**

The paper focuses on the classical task of Programming By Example (PBE): given some (input,output) pairs, the goal is to generate a program that "fits" these examples (producing the outputs when given the inputs), and also generalizes well to new inputs.
The paper evaluates mostly 7B and also 33B LLMs on three PBE tasks. The paper finds that finetuning these LLMs on these tasks further boosts their accuracy.
The paper also investigates out-of-distribution (OOD) generalization, and finds that OOD can be improved using a semi-supervised approach, where the model is given (input,output) pairs from the new domain (but not the desired program); then the LLM samples potential programs that solve the (input,output) pairs; if the program is correct (which can be validated) - it is added to the training set and the LLM is trained / finetuned again iteratively.

**Strengths:**

1. The paper is very clear and easy to follow, and it contains many examples that are visualized nicely.
1. The paper connects modern LLMs with the classical problem of Programming By Example (PBE)

**Weaknesses:**

1. Undefined, non-scientific, message - the title of the paper is "Is Programming by Example solved by LLMs?". This title leads the paper ("We investigate here the extent to which large language models pretrained on source code can solve PBE"), but I think that it's an undefined question. What does "solve" mean? By construction, and according to the "no free lunch" theorem, PBE can never be "solved". So "solving" PBE just depends on the difficulty of the questions. Even if we could define "solve PBE", how would you measure it? Is 90% considered "solved"? Is 80% "solved"? This problem is further expressed in L214: "absolute performance in LOGO remains poor" - 16% accuracy is not "poor" when you do not compare it to anything. Any accuracy number below 100% is considered as "unsolved" as any other number, and 100% is not possible on a hard enough dataset (because of "no free lunch").

1. Novelty - this is mostly an evaluation paper, that does not introduce any new approach or technique. Further, from the empirical evaluation, the answer to the question "Is Programming by Example solved by LLMs?" is, as expected, is "somewhat, but not quite": nothing in the empirical results was surprising or unusual: (a) finetuning LLMs on task-specific data works well; (b) semi-supervision on OOD data helps; (c) using Python as the output programming language works much better than the DSLs of classical work, because modern LLMs were trained on much more Python data than niche DSLs.

1. The OOD claim is a bit weak, because only in the relevant section it is said that "assuming we have access to problems drawn from the testing distribution" (without their labels, but these labels can be sampled and validated).

1. The paper compares its approach (a finetuned LLM) to classic symbolic, DSL-based (non-learning / non-neural) approaches several times throughout the paper, and speaks in favor of the LLM-based approach. This comparison to classic approaches is a bit of a strawman, since it is quite obvious that 33B LLMs are much more powerful than Flashfill (which is a paper from 2011) (Table 1).
The paper also mentions that:
>We also find that the resulting system can cover a broader scope of problems than classic symbolic methods, owing to the use of a Turing-complete language, which, at least theoretically, allows learning any computable function.

And I think that such claims completely miss the point: the reason that LLMs are better than classic symbolic methods is **not** the use Turing-complete languages. LLMs would have been better than classic symbolic methods even if the classic symbolic DSLs were turing-complete as well. The reason is that LLMs were trained on trillions of Python tokens.

6. Another trivial claim: in Section 4.2, the authors find that "posterior description length is more predictive than program size and prior description length". Simplifying the paper's claim, without using words from probability, basically says: the perplexity of the desired output sequence is predictive of its accuracy on downstream tasks. I think that this claim is quite trivial, and is very common in practice in LLM training: measuring perplexity on a validation set is usually closely correlated with success on downstream tasks.  Isn't this posterior iexactly what the model was *trained* to predict?

**Questions:**

1. Can the authors evaluate the baseline where the "program" is the LLM itself? That is, the LLM is trained/prompted to predict output to unseen inputs, without going through an explicit program. I am asking this specifically in light of Figure 4 - the examples there seem to be much more easy to solve directly with an LLM (in a few-shot prompting fashion, possibly with chain-of-thought), than to write an explicit program for.
1. In L96 the authors write: "we use samples from a generative model to train an inference network, but we do not further train the generative model itself" - what does this exactly mean? What kind of model is each of the "generative model" and "inference network"? Which of them is a pretrained LLM? And why not further training the generative model itself?
1. What exactly does the "Search Budget (Num Samples)" mean in the experimental section? Does that mean "accuracy@k" - sample $k$ different outputs, and consider the output as correct if *any* of these $k$ outputs is correct?
1. In Figure 3 - What temperature was used, and what other temperatures did the authors explore, for their finetuned model and for the baselines such as GPT-4? Since evaluation depends on sampling of up to 200 outputs, the temperature might have a drastic effect on the success of each model. With a proper tuning of temperature, the order of curves in Figure 3 might be different.

## Summary
Overall, the paper is not wrong and is presented nicely, but its novelty is limited, I'm not sure about the validity of some of the results such as Figure 3, and most of its conclusions are expected. I am thus voting for a borderline reject.

---

> ### Author Rebuttal · Authors · 2024-08-05
>
> Thank you for the thoughtful review. We really believe that our responses and new experiment can address your concerns, and hope that you will agree. Please see below.
>
> > empirical results was not surprising or unusual
>
> Papers in the past year find negative results for LLMs on PBE [1-3], and none try unusual PBE domains like our visual graphics programs. For example, our system beats the neurosymbolic LambdaBeam [3], and in LambdaBeam, they also compare to a strong LLM baseline - a Python fine-tuned LLM internal to Google - which was found to *not* do well on PBE. Our system also beats systems like DreamCoder [4] on unusual domains such as LOGO, where DreamCoder-like approaches are thought to excel. Our results should surprise the cited authors, and are unusual in that sense.
>
> [1] Shi et al. ‘24 ICLR [2] Rule et al. 2024 Nature Comms. [3] Shi et al. ‘23 NeurIPS [4] Ellis et al. ‘21 PLDI
>
> > does not introduce any new approach or technique
>
> A core part of the paper is a wake-sleep inspired algorithm for OOD generalization. Would you mind pointing us to a citation that we can refer to which previously introduced our specific approach/technique for adaptation?
>
> > The OOD claim is a bit weak... "assuming we have access to problems drawn from the testing distribution" (without their labels, but these labels can be sampled and validated).
>
> Deployed PBE systems are presented with a stream of "testing problems" generated by end users. In the wild, the requisite data is available for our adaptation technique.
>
> > What does "solve" mean?
>
> We’ll revise to make explicit our different definitions of solve:
> 1. Beating SOTA on mainstream benchmarks, including those used for deployed real world PBE systems [we solve in that sense]
> 2. Beating SOTA on unusual benchmarks very different from pretraining data [we solve in that sense: LOGO]
> 3. Achieving 100% success rate [we don’t solve in that sense, no free lunch]
> 4. Being practical for deployment [we don’t solve in that sense]
> 5. Generalizing out of distribution [partly: see our response above to "The OOD claim is a bit weak"]
>
> > “absolute performance in LOGO remains poor" - 16% accuracy is not "poor" when you do not compare it to anything
>
> We compare against 3 baselines (Fig 3)
>
> > Can the authors evaluate the baseline where the "program" is the LLM itself?
>
> Thanks for the suggestion. Please see below for the result of your proposed baseline, which does well on strings but not on lists.
>
> - Strings
>
> |Model|Accuracy|
> |--|--|
> |gpt-4-turbo (LLM as program)| 74.0%|
> |gpt-4-turbo (Python program)|73.9%|
> |deepseek-33b (LLM as program)| 49.0%|
> |deepseek-33b (Python program)| 70.9%|
> |ours-7b|76.9%|
> |ours-33b|81.2%|
>
> - Lists
>
> |Model|Accuracy|
> |--|--|
> |gpt-4-turbo (LLM as program)| 45.5%|
> |gpt-4-turbo (Python Program)|58.6%|
> |deepseek-33b (LLM as program)| 23.4%|
> |deepseek-33b (Python program)|46.2%|
> |ours-7b|75.8%|
> |ours-33b|79.0%|
>
> > Another trivial claim: in Section 4.2, the authors find that "posterior description length is more predictive than program size and prior description length"... basically says: the perplexity of the desired output sequence is predictive of its accuracy on downstream tasks
>
> Thank you for catching an error in the writing: By posterior we meant the marginal probability of solving the task under $q$ (marginalizing over sampled programs we found that pass), not evaluating perplexity of a single gold-truth target sequence. While it’s unsurprising this marginal is *at least* as predictive as the prior, it is noteworthy that it is *more* predictive than the prior, showing the LLM does not engage in blind guess-and-check, and did not merely learn to sample from the self-instruct distribution.
>
> > the reason that LLMs are better than classic symbolic methods is not the use Turing-complete languages... The reason is that LLMs were trained on trillions of Python tokens
>
> LLMs benefit from both massive data *and* expressive Turing-complete programming languages. As a thought experiment, consider an LLM trained on trillions of programs in the FlashFill++ DSL: It would still be utterly unable to solve the problems in Fig 4.
>
> > In L96 the authors write: "we use samples from a generative model to train an inference network, but we do not further train the generative model itself" - what does this exactly mean?
>
> We sample from the generative model $\mathcal{G}$, defined by prompting an LLM with seed problems (see L86). The inference network $q$ is our fine-tuned model (see L91). The prior $\mathcal{G}$ is a function of the seeds, so by holding the seeds fixed, we don’t update/train the generative model further, deferring such updates to Eq 5 (Adaptation).
>
> > What exactly does the "Search Budget (Num Samples)" mean in the experimental section?...  [Do you] sample different outputs, and consider the output as correct if any of these outputs is correct?
>
> Thanks for inviting this clarification. We model PBE in-the-wild where we commit to a single program and must run it on new unseen test inputs. This means we sample $k$ programs (the search budget), filter by the training input-outputs, pick a program randomly if more than one passes that filter, and finally report success only if that program correctly predicts all test input-outputs (following Eq 1). We will revise to clarify.
>
>
> > What temperature was used… Since evaluation depends on sampling of up to 200 outputs, the temperature might have a drastic effect on the success of each model
>
> The first sentence of the appendix gives the temperature ($T=1$). We used this somewhat high temperature (by the standards of code generation) because we wanted more diversity in the outputs, and did not tune this parameter because of the high cost of our experiments.
>
> Please let us know if you have further questions.

---

> ### Comment · Reviewer_mgYE · 2024-08-12
> **Response to authors**
>
> Thank you for your response.
>
> >Papers in the past year find negative results for LLMs on PBE [1-3]
> >and in LambdaBeam, they also compare to a strong LLM baseline - a Python fine-tuned LLM internal to Google - which was found to not do well on PBE
> >Our system also beats systems like DreamCoder [4]
>
> * LambdaBeam compared to PaLM 62B - PaLM is also ~2 years old, and as far as publicly known, pre-RLHF. I am not surprised that the most recent GPT-4 performs significantly better.
> * DreamCoder (2020), while being algorithmically clever, was also in the pre-LLM era.
> * I'm not sure what is the "Shi et al. ‘24 ICLR" - can you please give a link? I don't think it is cited in your paper.
> * I also cannot find a public link to "[2] Rule et al. 2024 Nature Comm" (although it is cited, I can't find a PDF).
>
> >A core part of the paper is a wake-sleep inspired algorithm for OOD generalization. Would you mind pointing us to a citation that we can refer to which previously introduced our specific approach/technique for adaptation?
>
> As I understand from the paper, the wake-sleep inspired algorithm is explained under standard "finetuning", and "adaptation" starts only later (in page 4). Is the wake-sleep inspired algorithm for finetuning or adaptation?
> Further, regarding the novelty of the wake-sleep algorithm, as mentioned in the paper:
> >This method is closely related to self-instruct [29] and wake-sleep [30]. Like self-instruct, we use
> prompting to bootstrap a large dataset from a small manually-constructed one. Our method differs
> by using the LLM to generate a hidden latent variable (the program) while a different generative
> process produces an observed variable (the program outputs).
>
> So, what is the difference between this paper and [29,30]? That a **different** model generates the program outputs, instead of being the same model?
>
> >Deployed PBE systems are presented with a stream of "testing problems" generated by end users. In the wild, the requisite data is available for our adaptation technique.
> It's a similar setting for a variety of machine-learning-based systems (e.g., Google Translate, Siri, etc.).
> If a system depends on "a stream of testing problems generated by end users", how does it respond to the first end-user queries? The authors's argument is almost like saying: "in a real system, we will wait for a few (hundreds? thousands?) queries to be sent, and only then we will be able to respond".
>
> In contrast, a machine learning system that "truly" generalizes OOD, should ideally generalize starting from the first OOD example.
>
> I know that it's OOD generalization starting from the first OOD example is not trivial and is still one of the open problems in machine learning - I just said that the OOD argument is a bit weak.
>
> >Beating SOTA on mainstream benchmarks, including those used for deployed real world PBE systems [we solve in that sense]
>
> I would still argue that beating SOTA does not mean that the problem is "solved".
>
> >>“absolute performance in LOGO remains poor" - 16% accuracy is not "poor" when you do not compare it to anything
>
> >We compare against 3 baselines (Fig 3)
>
> I don't understand this response - LOGO is "graphics", right? Which means Fig 3(c)? Isn't "Ours" better than the baselines there? So why is performance considered "poor"?
>
> > Please see below for the result of your proposed baseline, which does well on strings but not on lists.
> Thank you for these additional results.
>
> >LLMs benefit from both massive data and expressive Turing-complete programming languages. As a thought experiment, consider an LLM trained on trillions of programs in the FlashFill++ DSL: It would still be utterly unable to solve the problems in Fig 4.
>
> But would it change the downstream results if we improved any DSL to be Turing complete?
>
> >. We used this somewhat high temperature (by the standards of code generation) because we wanted more diversity in the outputs, and did not tune this parameter because of the high cost of our experiments.
>
> Tuning the best temperature for each model separately can significantly affect the results and the order between the models.
> This tuning is only needed at test time.

---

> > ### Author Response · Authors · 2024-08-13
> >
> > Thank you for your engagement. Our biggest disagreements concern the novelty of the methods and whether the empirical results are surprising. Please see below.
> >
> > ## **Methodological Novelty**
> >
> > > So, what is the difference between this paper and [self-instruct, wake-sleep]?
> >
> > Our adaptation algorithm is a novel hybrid of self-instruct and wake-sleep. Unlike self-instruct it can update its prior using new (unlabeled) data. Unlike wake-sleep it can adapt from few examples, via in-context learning. Mathematically this can be understood by looking at Equation 5 and making the following observations---which we’re revising to include right after Equation 5:
> >
> > > Equation 5 should be seen as a wake-sleep algorithm where "dreaming" corresponds to training $q$ on fantasy data (first equation) while "waking" corresponds to running inference and updating the prior $\mathcal{G}$ (by updating the seed, second pair of equations).
> >
> > Thank you for pushing us to clarify the writing concerning the novel conceptual aspects of our work. The revision will include the above two paragraphs.
> >
> > (Last, you should think of adaptation as wake-sleep: finetuning is like only having the sleep phase.)
> >
> > ## **Significance of Empirical Results**
> >
> > To avoid subjectivity, it’s helpful to consult the literature to understand what those in the area would find surprising/noteworthy/significant. Papers in the last year find negative results on LLMs for PBE:
> > 1. [ICLR ‘24, from DeepMind](https://arxiv.org/pdf/2307.13883) pg 7: **"LLMs in general perform poorly on program synthesis tasks specified only through I/O examples"**, *even for PaLM2 Unicorn, the largest PaLM2 model*
> > 2. [Nature Comms. ‘24, Rule et al](https://www.nature.com/articles/s41467-024-50966-x): see Fig 3
> > 3. [ICLR ‘24: Hypothesis Search](https://arxiv.org/abs/2309.05660) evaluates on ARC **finding GPT4 underperforms older symbolic solvers** (see [here](https://arxiv.org/abs/2402.03507) and [here](https://arxiv.org/pdf/2103.05823))
> > 4. [NeurIPS ‘23: LambdaBeam](https://arxiv.org/abs/2306.02049): As you point out this compares against a first-gen medium-size PaLM, but the above ICLR ‘24 paper compares against the latest-and-greatest PaLM2 Unicorn and arrives at similar conclusions.
> >
> > Our findings go beyond merely cheerleading LLMs. We instead show how to train small open models (7B, not RLHF’d) to surpass both bespoke neurosymbolic methods and massive closed-source systems, and also investigate uncommon creative applications such as visual graphics code. (Surprise is subjective, but we were *shocked* when LOGO graphics worked!)
> >
> > ## **Miscellaneous**
> >
> > > I just said that the OOD argument is a bit weak
> >
> > Although weakness/strength is subjective/relative, it’s helpful to consult the relevant literature for comparison. OOD for neural program synthesis has been previously studied using [architecture/prompt engineering](https://arxiv.org/pdf/2307.13883) or [handcrafted feature engineering](https://arxiv.org/pdf/1912.12345). Our (wake-sleep$\cap$self-instruct) algorithm instead does unsupervised domain adaptation, requiring **just tens of unlabeled OOD examples**. This is more generic and scalable than handcrafting features/prompts/architectures, hence a major strength of the approach relative to the prior art.
> >
> > > “absolute performance in LOGO remains poor" - 16% accuracy is not "poor" when you do not compare it to anything.
> >
> > The full quotation reads: "absolute performance in LOGO remains poor (*compare Fig. 6c to Fig. 3c*)". Contrasting Fig. 6c (OOD) to Fig. 3c (in-distribution) shows LOGO OOD performs poorly (but please see the updated LOGO results in the global response).
> >
> > > But would it change the downstream results if we improved any DSL to be Turing complete?
> >
> > Yes, because symbolic systems like FlashFill++ hinge on the clever design of non-Turing complete languages to judiciously restrict the search space. Going Turing-complete destroys the tractability of search and significantly impairs performance for (neuro)symbolic methods. (See [DreamCoder page 13 ](https://dl.acm.org/doi/pdf/10.1145/3453483.3454080): So intractable, search used a *year* of CPU time).
> >
> > > I would still argue that beating SOTA does not mean that the problem is "solved".
> >
> > Yes, hence why we propose five different subjective notions of “solve”, instead of merely declaring victory upon beating SOTA. The "scare quotes" were meant to emphasize the subjectivity of the quoted term, but we can revise to avoid "solve", including changing the title of the paper.
> >
> > ## **Parting Words**
> >
> > Obviously, we’re butting heads. But arguing these points has clarified to us how to communicate precisely what it is that is methodologically novel and empirically significant, and we’re confident that refactoring the writing can make that come through in the final paper. While we’d love to have your support, even if we don’t, thanks for forcing us to communicate these issues more clearly.

---

> > > ### Comment · Reviewer_mgYE · 2024-08-14
> > > **Response to authors**
> > >
> > > Thank you for the clarifications.
> > >
> > > I am not convinced by the novelty of the wake-sleep algorithm, and I do not think that this is a significant part of the paper.
> > > It could have been much easier for me to accept the paper if it was presented as an evaluation paper, without trying to claim novelty on the algorithm. I think that the paper can be more easily accepted by the audience as well if it was presented as an evaluation paper, so if I were the authors I would tone down the novelty of the algorithm.
> > >
> > > However as an evaluation paper, it is a good paper, and I hope that it will finally convince the rest of the program synthesis community to work with realistic LLMs and Python.
> > > I increased my rating from 4 to 6, good luck!

---

> > > > ### Author Response · Authors · 2024-08-14
> > > >
> > > > Thank you so much for your support and for your help in improving the paper; we will revise accordingly. We're very encouraged by your comments on the paper's potential impact on the program synthesis community!

---

### Official Review · Reviewer_e1ch · 2024-07-14

**Soundness:** 2
**Presentation:** 3
**Contribution:** 2
**Rating:** 5
**Confidence:** 3

**Summary:**

The paper performs a relatively thorough study on using LLM for example-guided program synthesis tasks. The results presented in the paper suggest that LLMs make strong progress toward solving the typical suite of example-guided synthesis tasks, potentially increasingly the flexibility and applicability of PBE systems.

**Strengths:**

- The PBE problem is interesting and well-motivated. Major papers in the field are well cited and referenced
- Extensive amount of traditional datasets are being evaluated
- The insights derived from experiments are somewhat valuable

**Weaknesses:**

- CoT and other simple prompting methods are not evaluated
- While there is an extensive amount of experiments and comparisons, we find that the outcome is relatively predictable.
- While the writing is generally okay and easy to understand, multiple typos and mistakes found in the writing (also mentioned in questions). Please consider fixing them.
- The LOGO visual examples are converted to an ASCII grid of characters (Fig. 8b). This might not be the most intuitive representation. Details about the transformation is not shown, such as how each number (0-9) is derived, the resolution of the ASCII grid, etc. With this design, it does not make sense for a non-fine-tuned LLM to solve the task. But technically you could still fine-tune GPT-3.5 with these inputs, but I guess it is okay to not include this experiment.

**Questions:**

- (Typo) line 150, there should be a space between (Tbl. 1,Fig. 3b)
- (Typo) figure 6a “Sygus” -> “SyGuS”
- (Grammar) last sentence of Figure 4 caption has grammar mistakes
- (Grammar) last sentence of Table 1 caption has grammar mistakes
- Appendix A.2 is empty
- I see in the prompt the authors wrote “You are a CS professor”. As far as I know this might not be the perfect prompt for code generation (this is just a joke).

---

> ### Author Rebuttal · Authors · 2024-08-05
>
> We thank Reviewer e1ch for the thoughtful review. Please see the global response PDF for the requested LOGO graphics details, and below for other new experiments and responses to your specific questions.
>
>
>
> > CoT and other simple prompting methods are not evaluated
>
> Thanks for the suggestion. We evaluated chain-of-thought and summarized the results below.
>
> - String Domain
>
> | Model           | Accuracy |
> | --------------- | -------- |
> | gpt-4-turbo with CoT | 76.5% |
> | gpt-4-turbo     |  73.9%      |
> | deepseek-33b |  70.6%  |
> | ours-7b | 76.9%|
> | ours-33b|**81.2%**|
>
> - List Domain
>
> | Model           | Accuracy |
> | --------------- | -------- |
> | gpt-4-turbo with CoT | 60.7% |
> | gpt-4-turbo     |  58.7%      |
> | deepseek-33b |  46.3%  |
> | ours-7b | 75.8%|
> | ours-33b|**79.0%**|
>
> We'll update the manuscript to include these results.
>
> > the ASCII grid [for LOGO], etc. With this design, it does not make sense for a non-fine-tuned LLM to solve the task. But technically you could still fine-tune GPT-3.5 with these inputs, but I guess it is okay to not include this experiment.
>
> Motivated by your observation we ran a new experiment using the multimodal GPT-4o and GPT-4o-mini with a few-shot prompt as a baseline for LOGO. This baseline was prompted with example programs and images and then given a new test image to write a program for. Please see below:
>
> |System |Method Type|Accuracy|
> |---|---|---|
> |Ours-33b  | LLM + Symbolic | 90%|
> |Ours-7b                               | LLM + Symbolic       | 89% |
> |GPT-4o (new result)                        | VLM + Symbolic    | 59%|
> |GPT-4o-mini (new result)                        | VLM + Symbolic    | 25%|
> |Regal (ICML’24)           |LLM + Library Learning | 57%|
>
> > The LOGO visual examples are converted to an ASCII grid of characters (Fig. 8b). This might not be the most intuitive representation. Details about the transformation is not shown, such as how each number (0-9) is derived, the resolution of the ASCII grid, etc.
>
> Please see the attached PDF, which shows an example of the ASCII transformation with a detailed conversion process in the caption. Interestingly, because the transformation down-samples the image, it is able to somewhat generalize to hand drawings, which we also show in the attached PDF, and would include in a revision.
>
> > Outcome is relatively predictable
>
> Papers in the past year find negative results for LLMs on PBE [1-3], and none try unusual PBE domains like our visual graphics programs. For example, our system beats the neurosymbolic LambdaBeam [3], and in LambdaBeam, they also compare to a strong LLM baseline - a Python fine-tuned LLM internal to Google - finding that the LLM is a poor PBE solver, counter to our findings. Our system also beats systems like DreamCoder [4] on unusual domains such as LOGO, where DreamCoder-like approaches are thought to excel. Our results are not predictable given the state of the recent literature.
>
> [1] Shi et al. ‘24 ICLR [2] Rule et al. ‘24 Nature Comms. [3] Shi et al. ‘23 NeurIPS [4] Ellis et al. ‘21 PLDI
>
> > Typos and grammar mistakes
>
> Fixed! Thank you for pointing them out.
>
> > I see in the prompt the authors wrote “You are a CS professor”. As far as I know this might not be the perfect prompt for code generation (this is just a joke).
>
> We will consider adding to the prompt “You are reviewer #2, please improve the code like how you would improve a paper” (this is just a joke).
>
> Thanks again for the review and please let us know if we can answer any further questions.

---

### Official Review · Reviewer_CppC · 2024-07-18

**Soundness:** 3
**Presentation:** 3
**Contribution:** 3
**Rating:** 7
**Confidence:** 3

**Summary:**

This paper investigates whether the long-studied programming by example task is "solved" by large language models with Turing-complete languages like python.
Their evaluation is on three domains: lists, strings, and LOGO/Turtle graphics.
They evaluate three LLM-based approaches, including a self-instruct-like fine-tuning approach that tunes LLMs on synthetic labeled data, and an adaption approach assuming access to problems (not solutions) from the testing distribution.
Compared to several symbolic, neurosymbolic, and LLM baselines, the proposed approaches perform better.
The analysis of the correlation between different aspects of the target program indicates that the fine-tuned model is beyond blind guess-and-check.

**Strengths:**

1. The experiments are comprehensive, and the analysis of different predictors of model performance is helpful in understanding the extent to which LLMs solve PBE.
2. The proposed methods make use of the fact that PBEs problems can be accurately synthesized using model-generated inputs and programs. The experiment results show that they are effective in solving in-domain problems and adapting out-of-distribution ones at test time.
3. This paper answers some interesting questions regarding the role of LLMs for PBE and points out what researchers might work on in the future.

**Weaknesses:**

Contamination. As the authors acknowledged on Line 148, the problems could be in LLMs' pertaining data. I wonder if the authors have an idea of how much of a role such potential contamination plays in LLMs' superior performance. Is there anyway to rule out or minimize the impact of that confounder?

**Questions:**

1. How much does a turing-complete language help in solving PBE, excluding the fact that LLMs have seen lots of python code? Is the expressiveness of a turing-complete language itself helpful?
2. How far can the adaption go? Right now the adaption discussed is still within the same category of problems (such as lists), I imagine a more general PBE system might be able to adapt to problems that are more different.

**Limitations:**

Yes.

---

> ### Author Rebuttal · Authors · 2024-08-05
>
> Thank you for the detailed review, and for your support. Please see the global review for a PDF with new results (including a fun new LOGO experiment). We address your specific questions below.
>
> > Regarding Contamination
>
> We avoided contamination as follows:
> 1. String dataset: The datasets contain only input/output examples, and thus the program could not be in pretraining data.
> 2. List dataset: The dataset from Rule et al. does not include program solutions, and furthermore, the dataset is in BigBench, which is conventionally excluded from LLM pretraining data.
> 3. LOGO dataset: The python program LOGO dataset (from Regal) was released within 1 week of the beginning of training of deepseek (the model that we fine tune), so is almost certainly not in pretraining. Just in case, we further tried prompting the model to confirm that it cannot complete a partial ground truth python LOGO program.
>
> We also tried running our system on new problems that we created by hand, illustrated in Figure 4, and also illustrated in the attached main-response PDF, where we ran the LOGO synthesizer on a hand drawing we made. The system was surprisingly able to handle all of these new problems that could not possibly have been in the pretraining data.
>
> > How much does a turing-complete language help in solving PBE, excluding the fact that LLMs have seen lots of python code? Is the expressiveness of a turing-complete language itself helpful?
>
> We believe the expressiveness itself is helpful due to the fact that a handcrafted restricted programming language may not be able to capture the space of all user-desired programs, even within a single domain. For example, in Fig 4 (top), we can see that a simple example—numbering the lines of a paragraph—can easily be solved by our model but cannot be solved by FlashFill++. We believe this is a general problem with handcrafted domain-specific programming languages: Although they make the search space smaller and therefore more tractable, they inevitably exclude important computations.
>
> In other words, LLMs benefit from both massive data *and* expressive Turing-complete programming languages. As a thought experiment, consider an LLM trained on trillions of programs in the FlashFill++ DSL: It would still be utterly unable to solve the problems in Fig 4.
>
> > How far can the adaption go? Right now the adaption discussed is still within the same category of problems (such as lists), I imagine a more general PBE system might be able to adapt to problems that are more different
>
> Adaptation hinges on solving at least some problems in the OOD target, and we have observed that after fine-tuning for domain A, the model can still solve some problems in domain B. We believe this is because LoRA finetuning does not cause *too* much catastrophic forgetting about domain B.
>
> We’ll update to include experiments of cross domain adaptation (lists task adapt to string task, string task adapt to list task), which could be very interesting as progress toward truly general purpose program synthesis. Thanks for the suggestion!

---

### Author Rebuttal · Authors · 2024-08-06

Thank you all for the helpful reviews. Please see your individual responses, but here we wish to include a PDF illustrating:
1. The conversion to ASCII art requested by reviewer e1ch. Interestingly, we also found that by down sampling the image to ASCII, it is able to somewhat generalize to hand drawings (also shown).
2. New baselines including a multimodal model for graphics programming (GPT-4o and GPT-4o-mini), chain-of-thought prompts, and others. (Our LOGO results are now higher than in the original submission because we fixed a small programming issue that was degrading performance)

---

### Decision · Program_Chairs · 2024-09-25

**Decision:**

Accept (poster)

**Comment:**

This paper presents an evaluation of a variety of programming by example (PBE) by LLMs and "traditional" symbolic approaches. This work finds that off-the-shelf pretrained LLMs are not effective but can be tuned to achieve acceptable performance. This work further showcases a method for helping LLMs solve OoD with a "lightweight" adaptation.

Overall, this work considers an interesting problem and presents a good evaluation of different techniques for PBE and shows gaps in many LLMs. Since PBE requires core reasoning skills it shows gaps in existing systems. While there are a few additional evaluations that have been asked by the reviewers (which would be nice to see in a camera-ready), I believe that it would be valuable to publish this paper at this time.


### Minor
* Self-repair (ie multi-turn with execution feedback) is not included in the baselines and I believe it would be useful to compare to.
* A better discussion of (inference) cost vs accuracy is needed. If a symbolic system was given as much budget as an LLM needs (CPU-dollars vs GPU-dollars) to which extent would the results change?